# Hybrid PET–MRI Imaging in Paediatric and TYA Brain Tumours: Clinical Applications and Challenges

**DOI:** 10.3390/jpm10040218

**Published:** 2020-11-09

**Authors:** Ananth Shankar, Jamshed Bomanji, Harpreet Hyare

**Affiliations:** 1Children and Young People’s Cancer Services, University College London hospitals NHS Foundation Trust, London NW1 2PG, UK; 2Department of Nuclear Medicine, University College London hospitals NHS Foundation Trust, London NW1 2PG, UK; jamshed.bomanji@nhs.net; 3Department of Radiology, University College London Hospitals NHS Foundation Trust, London NW1 2PG, UK; harpreet.hyare@nhs.net; 4Department of Brain Repair and Rehabilitation, Institute of Neurology, University College London, London WC1N 3BG, UK

**Keywords:** PET, MRI, glioma

## Abstract

(1) Background: Standard magnetic resonance imaging (MRI) remains the gold standard for brain tumour imaging in paediatric and teenage and young adult (TYA) patients. Combining positron emission tomography (PET) with MRI offers an opportunity to improve diagnostic accuracy. (2) Method: Our single-centre experience of ^18^F-fluorocholine (FCho) and ^18^fluoro-L-phenylalanine (FDOPA) PET–MRI in paediatric/TYA neuro-oncology patients is presented. (3) Results: Hybrid PET–MRI shows promise in the evaluation of gliomas and germ cell tumours in (i) assessing early treatment response and (ii) discriminating tumour from treatment-related changes. (4) Conclusions: Combined PET–MRI shows promise for improved diagnostic and therapeutic assessment in paediatric and TYA brain tumours.

## 1. Introduction

Tumours of the Central Nervous System (CNS) are the most common form of solid tumour in children and teenage/young adult (TYA) populations, accounting for more than 25% of all types of cancers and 15% of malignancies in the teenage and young adult population (15–24 years). According to the most recent statistics from Cancer Research UK, tumours of the CNS are the fourth most common cancer type in this age group with astrocytomas forming the largest sub-group followed by embryonal tumours, such as medulloblastoma and ependymoma [1,2]. Treatment is dependent upon histological subtype, tumour molecular genetic profile and staging, with varying combinations of surgery, radiotherapy and chemotherapy. Recent advances in molecular characterization of CNS tumours provide additional information that allow for risk adapted tailored treatment [3,4,5].

Imaging plays a key role in the management of paediatric/TYA patients with brain tumours at all stages of the diseases: from diagnosis, surgical and radiotherapy planning to the assessment of therapeutic response and detection of recurrent disease. MRI, with its excellent soft tissue contrast, remains the imaging modality of choice in daily practice. However, interpretation of standard anatomical MRI after surgical trauma, disrupting normal CNS anatomy, and treatment with chemoradiation, inducing oedema, necrosis, demyelination and ischaemia, can be challenging. The recently characterised phenomena of *pseudoprogression*, whereby an increase in tumour volume, oedema and enhancement shortly after completion of treatment is often difficult to distinguish from progressive tumour [6,7] and *pseudoresponse,* whereby a dramatic reduction in tumour enhancement following treatment with anti-angiogenic agents is thought to be due to vascular normalisation rather than a true anti-tumour effect [8,9] further compound interpretation. With the recent evidence that non-enhancing WHO grade II and III astrocytomas lacking the IDH1 mutation should be classified as glioblastoma (GBM) [10] and that not all high-grade gliomas enhance [11], there is a need to investigate other imaging technologies to better assess tumour burden.

Various functional imaging techniques depicting specific biological properties of tissues have been introduced to compliment standard MRI, including advanced MRI techniques and positron emission tomography (PET). Each modality is reported as demonstrating increased diagnostic accuracy over standard MRI in neuro-oncology and combining PET with advanced MRI has the potential to further improve diagnostic accuracy and ultimately patient outcomes.

Whilst the role of ^18^F-fluoro-deoxy-glucose (FDG) PET/CT is well established in oncology, the clinical role of PET–MRI, first introduced in 2008 [12], has yet to be established. Recently, there have been a number of reports exploring the potential of hybrid PET–MRI in neuro-oncology [13,14,15], but this has been limited to adult patients. Within paediatric and TYA patients, the reported use of hybrid PET–MRI consists of small feasibility studies and case reports [16,17,18]. Here, we present a critical appraisal of the clinical use of PET–MRI in a group of paediatric/TYA neuro-oncology patients based on over 5 years of experience predominantly using ^18^F-fluorocholine (FCho) and ^18^fluoro-L-phenylalanine (FDOPA) on our Siemens mMR system, which performs over 1200 PET–MRI research and clinical studies each year. We present (i) an overview of the current imaging techniques used in clinical practice, (ii) a series of clinical cases at our institution, highlighting the challenges and opportunities of hybrid PET–MRI, and (iii) finally we present recommendations for clinical use on paediatric and TYA neuro-oncology and some limitations.

## 2. Current Clinical Imaging Methods in Neuro-Oncology

### 2.1. PET Tracers

Positron emission tomography (PET) is an established imaging modality used commonly in oncology [19]. In the neuro-oncology domain, it is being increasingly used to add incremental value to MRI in the clinical management of glioma [20,21,22]. The potential to use PET imaging in clinical trials and new strategies for the treatment of glioma, such as immunotherapy, where pseudoprogression is particularly challenging for MRI assessment [23], remains to be explored. An evidence-based recommendation by the PET Response Assessment in Neuro-Oncology (RANO) working group and European Association of Neuro-Oncology (EANO) on the clinical use of PET imaging in gliomas has been published focussing on radiotracers used in clinical practice imaging, i.e., glucose metabolism, 2-deoxy-2-[^18^ F]fluoro-D-glucose (FDG), and system L amino acid transport comprising ^11^C-methyl-methionine (MET), O-(2-[^18^F]-fluoroethyl)-L-tyrosine (FET) and 3,4-dihydroxy-6-[^18^F]-fluoro-L-phenylalanine (FDOPA) [20].

Radiolabelled amino acids have been used in neuro-oncological practice since 1983, and whilst most experience has been gained with MET, radiolabelling with the isotope carbon-11 requires an on-site cyclotron unit. Labelling with the isotope fluorine-18, with a longer half-life, has advantages in clinical practice, and FET is now widely used across Europe with numerous studies reporting increased sensitivity and specificity for differentiating tumour progression from pseudoprogression [24,25,26]. FDOPA, which was developed to measure dopamine synthesis in the basal ganglia, is increasingly used for brain tumour imaging [21]. Amino acid PET tracers can penetrate the blood–brain or –tumour barrier (BBB/BTB) and thus provide excellent definition of tumours, including those that are non-contrast enhancing [27]. Their uptake is mediated via amino acid transporters LAT1 and LAT 2 [28,29].

Our initial experience has been with FCho, which showed promise as a diagnostic imaging tool in benign and malignant brain tumours [30,31,32]. Its effectiveness is related to its ability to indirectly evaluate cell proliferation as a measure of the synthesis of lipids required for cell membranes. Additionally, slow-growing tumours such the low-grade gliomas also take up radiolabelled choline. An important feature of radio-labelled choline is its low rate of distribution in normal grey and white matter [33,34]. Our clinical practice is now based on the amino acid tracer FDOPA, which demonstrates superiority over FCho in its ability to provide information on the metabolism of brain tumours, its ability to discriminate low-grade from high-grade gliomas, and the significant impact it has on the management of patients with a suspicion of brain tumour recurrence and the challenging clinical problem of distinguishing pseudoprogression from true progression [35,36].

Other amino acid PET tracers such as α-[^11^C]-methyl-L-tryptophan (AMT) and [18F]fluciclovie (FACBC) as well as glutamine-based amino acid PET tracers have also been evaluated in gliomas for tumour delineation, prognostication and differentiation of tumour recurrence from radiation injury, but currently, the number of examined subjects is low [37,38].

Promising new PET tracers include ^18^F-fluoromisonidazole (FMISO) for imaging hypoxia in brain tumours [39,40]; ^11^C-(R)PK11195, a translocator protein (TSPO) ligand which demonstrates increased binding in high-grade gliomas [41]; and ^68^Ga-FAPI, which is a labelled fibroblast activating protein inhibitor (FAPI) that binds to cancer associated fibroblasts (FAP) [42]. We plan to evaluate ^68^Ga-FAPI in our paediatric/TYA glioma population for the monitoring of treatment response.

### 2.2. MRI

Standard basic minimal anatomical MRI for paediatric brain tumour imaging includes axial fluid-attenuated inversion recovery (FLAIR) and T2-weighted sequences visualizing non-enhancing tumour components, oedema and therapy induced-gliosis, diffusion weighted imaging (DWI) to aid the assessment of therapy response and high-resolution 3D isotropic pre- and post-contrast enhanced T1-weighted sequences to depict contrast leakage from the tumour vessels (Figure 1) [43]. Whilst international standard brain MRI protocols exist for adults [44], no such recommendations exist for paediatric neuro-oncology [45]. At our institution, we follow the adult MRI recommendations but also include a blood sensitive sequence such as T2* or susceptibility weighted imaging (SWI) to depict intra-tumoural haemorrhage and the development of radiation-induced cavernomas, and to depict biopsy tracts [46] and arterial spin labelling (ASL) perfusion to aid in the assessment of gliomas post-treatment [47]. Table 1 demonstrates the parameters for our paediatric/TYA neuro-oncology MRI protocol.

Pre- and post-contrast T1-weighted sequences using identical MRI parameters allow voxel-to-voxel image subtraction, which may detect subtle residual enhancement and therefore provide a more accurate and reproducible assessment of the true extent [48]. Whilst this method has been shown to significantly improve visualization and quantification of post-treatment high-grade glioma tumour volume in adults [49], this method needs further validation in paediatric/TYA gliomas.

A variation of this method acquires two high-resolution 3D T1-weighted sequences in the same MRI session, and 3–5 and 60–75 min after the injection of the contrast medium and subsequent subtraction a colour-coded map known as a treatment response assessment map (TRAM) is produced. Areas where contrast accumulates correspond histologically to treatment-induced changes, whilst areas where contrast is cleared rapidly correspond to active tumour [50]. TRAMs are simple to acquire but can be inconvenient to the patient who may have to wait longer to complete the study, a particularly challenging in the paediatric population.

#### 2.2.1. Diffusion-Weighted Imaging (DWI)

Diffusion-weighted imaging (DWI) is sensitive to random microscopic (Brownian) motion of water molecules, which results in signal loss and consequent hyperintensity in areas of restricted diffusion. The apparent diffusion coefficient (ADC) reflects the magnitude of water motion with restricted diffusion having lower ADC values. ADC is inversely correlated with cell density [51]due to reduced water mobility from dense cellular packing and thus is of interest for grading tumours and detection of tumour recurrence. In paediatric neuro-oncology, ADC has been shown to correlate with tumour grade, and in a meta-analysis that included 14 studies of paediatric patients, lower-mean tumour ADC was associated with a higher WHO tumour grade, except for diffuse midline gliomas which demonstrated ADC characteristics similar to low-grade gliomas [52].

One caveat to the application of DWI for identifying non-enhancing tumour is that lower ADC values do not always correlate with increasing tumour infiltration. Low and persistent diffusion restriction can be associated with non-viable tissue necrosis. In adults, these lesions tend to be periventricular, slowly change over several months and are thought to be a type of treatment toxicity, associated with better outcomes. Novel diffusion modelling techniques such as Vascular, Extracellular, and Restricted Diffusion for Cytometry in Tumours (VERDICT) have been developed at our institution that aim to identify diffusion parameters that more accurately reflect tumour microstructure [53] and will be applied to paediatric brain tumours (Figure 2).

#### 2.2.2. Perfusion-Weighted Imaging (PWI)

Perfusion-weighted imaging (PWI) exploits the neoangiogenic properties of proliferating gliomas, and is able to identify areas of high-grade tumour with high accuracy. PWI can be performed in a number of ways; the main techniques in clinical practice are dynamic susceptibility-weighted contrast-enhanced MRI (DSC), dynamic contrast-enhanced MRI (DCE) and arterial spin labelling (ASL).

DSC-MRI rapidly acquires gradient echo or spin echo planar images during first-pass transit through the brain of an exogenous paramagnetic gadolinium based-contrast agent that transiently decreases signal intensity. Voxel-wise changes in contrast agent concentration are determined from time–signal curves and processed using kinetic tracer modelling and indicator dilution theory to estimate cerebral blood volume (CBV), cerebral blood flow (CBF) and mean transit time (MTT). Relative CBV (rCBV) is the most common DSC-MRI metric for evaluating brain tumours. To minimize the error caused by contrast extravasation, known to occur in brain tumours, preloading of the contrast agent along with model-based post-processing leakage correction can decrease both T1 and T2* effects [54,55].

DCE-MRI is based on the T1 relaxivity properties of a contrast agent and quantifies various dynamic features of blood–brain barrier contrast agent leakage. After baseline T1 maps are obtained, T1-W DCE-MRI images are acquired before, during and after contrast agent administration. A vascular input function is determined, and pharmacokinetic modelling yields the extravascular extracellular and plasma space volume fractions, transfer constant (K^trans^) and rate constant. K^trans^ is thought to reflect microvascular permeability but also representative of blood flow and vessel surface area, the most common metric in brain tumour studies [56].

ASL uses endogenous contrast whereby magnetically labelled blood entering the brain can be used to estimate the CBF [57]. The first applications used a long RF pulse simultaneously with a selection gradient, but more recently, a train of short RF pulses combined with a strong gradient has been used (pCASL) [58]. Although, DSC perfusion is most commonly used in clinical practice [59] and numerous studies have demonstrated that rCBV is an early response marker in treated GBMs, pCASL is the favoured PWI technique in paediatric neuro-oncology due to the lack of need for exogenous contrast [60]. pCASL has been shown to correlate with DSC-derived CBF in a study of 15 paediatric gliomas, and numerous studies have demonstrated that increased pCASL-derived CBF is associated with a higher tumour grade [47,52]. pCASL is the preferred PWI technique used at our institution and is part of the neuro-oncology MRI protocol for assessment of treated gliomas (Figure 1). The role of pCASL in the evaluation of treated paediatric gliomas is yet to be established but is the focus of active research.

### 2.3. Hybrid PET–MRI

Although combining PET imaging with advanced MRI provides an opportunity to improve overall diagnostic accuracy in neuro-oncology, to date, very few published studies have investigated this potential, mainly focussing on feasibility or agreement of modalities [13,15]. In paediatric neuro-oncology, the literature is even more scarce, limited to case reports and small feasibility studies, but the prevailing literature does support a role for amino acid PET [20]. We have previously shown that 18F-choline PET–MRI improves response assessment in treated gliomas [17,30] and can be used to validate novel MRI techniques such as Amide Proton Transfer-Chemical Exchange Saturation Transfer (APT-CEST) [18] (Figure 3). In addition, we have shown that 18FDOPA PET–MRI improves assessment of post-treatment glioma burden compared to MRI alone [61].

## 3. Clinical Case Studies of Hybrid PET–MRI in Neuro-Oncology

Nevertheless, in paediatric neuro-oncology, hybrid PET–MRI is not routinely used in clinical practice and should be considered as research. At our institution, PET–MRI is a supplementary diagnostic aid performed after an equivocal MRI. In this next section, we present illustrative examples of our experience of PET–MRI in specific tumour types at different timelines during their treatment. For clarity, we have grouped them into 4 groups: high-grade glioma (HGG), low-grade glioma (LGG), intracranial germ cell tumours (ICGCT) and primitive neuroectodermal tumours (PNET):

### 3.1. High-Grade Gliomas

High-grade gliomas are a challenging group of tumours as they invariably have dismal treatment outcome. While combining the information obtained from conventional MR imaging with the ADC values could increase the accuracy of pre-operative differentiation between low-grade and high-grade paediatric tumours, this is not always possible. Additionally, differentiating pseudoprogression from true progression remains a diagnostic dilemma, as conventional magnetic resonance imaging (MRI) features often cannot discriminate pseudoprogression from early true progression after chemoradiotherapy as contrast-enhancing lesions on MRI that do not reflect true tumour progression.



**Patient 1: Diagnostic Biopsy and Radiotherapy Planning**



A 16-year-old male presented with a short history of sharp bilateral eye pain that was intermittent in nature. He was seen by an ophthalmologist who detected bilateral swollen optic discs suggestive of papilledema. A diagnostic FCho PET–MRI scan showed an expansile pontine tumour with heterogenous, predominantly high T2-weighted signal abnormality. Choline uptake was seen in the enhancing component (Figure 4). A neuro-navigational stealth-guided biopsy was performed, centred on the choline avid region of the tumour, and the histology confirmed this to be a high-grade astrocytic tumour with a very high proliferative index. A response assessment scan after 6 weeks of chemoradiotherapy showed excellent treatment response with a complete metabolic response.



**Patient 2: Early Response Assessment**



A 13-year-old female with a large tumour involving left thalamus and basal ganglia, infiltrating into the adjacent white matter, insular cortex and left aspect of the upper brain stem underwent a biopsy of the thalamic component, which confirmed high-grade astrocytoma with a mutation of the variant H3.3 gene. An early response assessment scan 6 weeks after completion of chemoradiotherapy demonstrated no residual enhancement in the tumour, but FDOPA PET showed residual tracer distribution consistent with residual metabolically active disease (Figure 5).



**Patient 3: Tumour Progression Versus Pseudoprogression**



A 5-year-old female with diffuse midline glioma (DMG) presented with a short history of headaches, difficulty in walking and double vision. Radiology at diagnosis was consistent with diffuse intrinsic pontine glioma (DIPG). She commenced on 6 week course of focal radiotherapy (RT) to the pontine tumour with good initial response (Figure 6). The patient then went to Mexico for a form of targeted treatment and after four cycles developed clinical signs of disease progression. Whilst follow-up MRI demonstrated increased enhancement and raised the possibility of pseudoprogression, FDOPA PET demonstrated areas of avidity confirming clear progression.



**Patient 4: Tumour Progression Versus Pseudoprogression**



A 15-year-old male with a large tumour involving the left thalamus underwent gross resection of the thalamic tumour, and the histology confirmed this to be a high-grade astrocytoma with the known histone H3F3A K27M mutation. A response assessment scan after 6 months of chemotherapy (Figure 7) suggested minimal enhancement in the residual tumour that was thought to be pseudoprogression. However, FDOPA showed tracer uptake consistent with tumour progression.



**Patient 5: End of Treatment Assessment**



An 18-year-old female presented with a short history of headaches, vomiting and blurred vision. MRI scan at diagnosis (Figure 8) showed a large temporal lobe tumour. She underwent a sub-total resection and then commenced a 6-week course of chemoradiotherapy followed by 14 months of maintenance chemotherapy. Whilst the gadolinium contrast MRI scan at the end of treatment 18 months later demonstrated residual enhancement, FCho demonstrated no uptake, suggesting that the mass was non-viable.

### 3.2. Low-Grade Gliomas



**Patient 6: Tumour Progression Versus Pseudoprogression**



A 10-year-old presented with low-grade glioma with a background of NF1 (neurofibromatosis type 1). A mid treatment MRI scan following disease progression showed enlargement of an enhancing lesion in the right cerebellar hemisphere, but the patient was well and clinically stable, and the radiological changes were thought to reflect pseudoprogression. An FDOPA PET–MRI showed increased tracer activity in the enhancing lesion along with a second FDOPA avid lesion in the right subthalamic region, consistent with metabolically active disease (Figure 9).



**Patient 7: Treatment Response**



A 7-year-old female presented with tremors and an upward gaze palsy. A MRI scan showed a large tectal plate tumour involving the right thalamus causing secondary hydrocephalus.

She had a ventriculo-peritoneal (VP) shunt inserted for CSF diversion, but 4 years later following tumour progression, she underwent a biopsy which confirmed low-grade glioma. FDOPA showed intense uptake in both the enhancing and non-enhancing components of the tumour (Figure 10), following which the patient was referred for proton beam treatment PBT]. Post PBT FDOPA at 8 weeks showed complete metabolic response.



**Patient 8: Suspicion of Transformation**



A 15-year-old female with known low-grade glioma during routine surveillance was noted to have developed a large lesion in the corpus callosum. This occurred 7 years after her initial diagnosis of an optic pathway tumour. The MRI scan raised the possibility of malignant transformation to high-grade glioma. FDOPA (Figure 11) discounted the possibility of transformation as the FDOPA uptake within the corpus callosal lesion was low intensity, unlike that seen in high-grade gliomas.

### 3.3. Intracranial Germ Cell Tumour

Intracranial germ cell tumours (ICGCTs) are uncommon tumours occurring in children and young adults. They are sub-categorised into germinomas (GGCT) and non-germinomatous tumours (NGGCT). While GGCT are highly curable tumours with multimodality treatment, NGGCTs have a poorer outcome. There are clear differences in the approach to the management of ICGCTs. Current treatment strategies focuses on reducing treatment intensity, particularly the dose and volume of radiotherapy, in order to minimise the risks of late sequelae while maintaining high cure rates. In addition, especially in those with NGGCT, survival outcomes are unsatisfactory, and complete surgical resection combined with more intensive chemotherapy and radiotherapy remains the best available treatment option at this time. Occasionally, second-look surgery is required when the residual tumour increases or remains stable, despite normalized or nearly normalized tumour markers, in order to achieve complete resection and improve outcome.



**Patient 9: Response Assessment**



A 19-year-old male presented with a short history of headaches and double vision. MRI of the brain and spine showed a multi-cystic tumour pineal region tumour causing obstructive hydrocephalus. As serum AFP and bhCG were elevated, a diagnosis of intracranial NGGCT was made. After four cycles of chemotherapy, 18F-fluoroethylcholine (FEC) PET–MRI showed stable tumour volume with moderate FEC uptake, suggestive of viable tumour (Figure 12). He underwent macroscopic tumour resection that histologically showed the presence of viable tumour and subsequently proceeded to cranio-spinal radiotherapy (CSRT) with a boost to the pineal tumour.

### 3.4. Primitive Neuro-Ectodermal Tumours (PNET)

Combining anatomical and functional imaging can improve sensitivity and accuracy of tumour diagnosis in high-grade paediatric embryonal tumours. Although PET using radiolabelled amino acids provides important metabolic information for astrocytic tumours, much less is known of the value of PET in medulloblastomas.



**Patient 10: Diagnosis**



This 15-year-old female had a short-presenting history of recurrent headaches, vertigo, back ache and a band like numbness around the right breast and chest. She also complained of her legs “giving away”. MRI of the brain and spine revealed multiple lesions involving the brain with multiple drop spinal lesions occupying the spinal canal, compressing the spinal cord and nerve roots at all the levels (Figure 13), suggestive of a high-grade tumour. Surprisingly, the FCho PET–MRI showed that all the lesions in the brain and spine were choline non-avid, suggestive of a low-grade tumour, i.e., a low proliferative index, but a subsequent biopsy of the spinal lesion confirmed medulloblastoma.

## 4. Discussion

Imaging is a critical component in the management of brain tumours, with MRI being the modality of choice. While conventional cross-sectional contrast-enhanced MRI provides structural information with regard to tumour size and location, it does not provide complimentary information on tumour metabolism, biological aggressiveness and prognosis. Various functional imaging have been developed that provide information on tissue properties and tumour metabolism, including advanced MR and PET. Combining the techniques in hybrid PET–MRI provides the opportunity to demonstrate structural and metabolic information, improving diagnosis. We have shown that the use of the hybrid PET–MRI imaging platform in our paediatric/TYA neuro-oncology practice has had a decisive bearing on treatment decisions, prognosis and overall outcome.

We have previously shown unequivocal evidence that FDOPA PET and FCho PET have high diagnostic accuracy for the detection of low- and high-grade gliomas and for intracranial germ cell tumours [17,30,31]. In gliomas, while the discrimination between tumour recurrence and radiation-induced changes has been well described [17], the parameters for treatment response assessment need further investigation with standardized imaging protocols in prospective studies.

### 4.1. FCho

We have previously shown that FCho PET–MRI is a promising and reliable imaging tool for children with astrocytic tumours, as it permits monitoring of morphological and metabolic response and changes during therapy [17,30]. FCho has been shown to be highly effective in confirming the absence of viable residual tumour, as illustrated in patient 5, or the presence of viable residual tumour (patient 9), confirming the predictive value of discrimination of viable from non-viable tumour. In addition, the use of PET is particularly useful to target the malignant foci for biopsy sites in gliomas, which is well illustrated by the case of patient 1 and in the management of patients with recurrent disease when combined with MRI [32]. There are also reports in the literature where PET has been used to improve the diagnostic accuracy [61] and radiation therapy planning in patients with gliomas [20,33].

FCho has been used as an imaging tool for diagnosis, assessment of treatment response and remission status for children and adolescents with intracranial germ cell tumours [31]. Management of intracranial germ cell tumours in TYA patients is complex, as many patients have an incomplete radiological response to standard treatment. These residual radiological lesions could harbour viable tumour, and, hence, surgical resection is recommended to improve survival outcome. In this series report, patient 9 demonstrated moderate FEC uptake, suggestive of residual viable tumour, enabling a decision to proceed with surgical resection that confirmed the presence of residual viable tumour. There were two further patients who had residual pineal lesions on MRI but were FCho PET negative. Both patients underwent surgical resection, and there was complete concordance with choline uptake and histology at resection: both had necrotic non-viable tumour at resection. We have not included these cases here as the scans were not performed on the hybrid PET–MRI scanner, but further details can be found in a previous publication [31].

### 4.2. FDOPA

Reports in published literature have highlighted the value of FDOPA PET as an imaging tool for early response assessment for high-grade gliomas [62,63]. The ability of FDOPA to better delineate treatment response was illustrated in patients 2 and 5 and differentiation of progression from pseudoprogression are well illustrated in patients 4 and 6. In patients 4 and 6, while treatment-related changes/pseudoprogression could not be excluded on the contrasted MRI, FDOPA was highly discriminatory, and the changes were consistent for true disease progression. While many paediatric low-grade gliomas unlike in adults are contrast-enhancing tumours, they are often heterogenous and do not enhance uniformly. Additionally, contrast enhancement resulting from increased permeability of the blood–brain barrier is nonspecific and does not demarcate tumour extent or treatment effect. Planning treatment, especially radiotherapy, can be difficult in such a situation. The use of a PET tracer such as 18F DOPA can better identify the non-enhancing tumour, as the increased uptake of FDOPA in gliomas appears to be related to increased transport via the amino acid transport system [28,29].

### 4.3. False Negative Studies

It must be emphasised that not all gliomas or embryonal tumours of the brain are FCho or FDOPA avid. It has been previously shown that while a significant correlation between FDOPA uptake and tumour proliferation in newly diagnosed tumours is usually observed, this correlation is not seen in recurrent tumours [64,65]. The reason attributed for this discordance is because recurrent tumours could have differential blood–brain barrier breakdown due to previous treatments. Therefore, it is not surprising that the correlation between ^18^FDOPA PET uptake and tumour grade is better seen in treatment-naïve newly diagnosed tumours. However, our patient had no previous treatment other than resection of her spinal astrocytoma at first diagnosis, thus, the latter explanation cannot be applied here. It is therefore important that a negative scan during any phase of treatment cannot be equated to complete metabolic response if there was no previous positive PET scans for comparison.

In our series, FCho demonstrated no uptake in patient 10 with disseminated medulloblastoma at diagnosis. Typically, medulloblastomas are hypermetabolic with a high proliferative index and have been shown to be choline avid [66]. Hence, the lack of choline avidity was surprising, and we can only speculate that this was due to technical failure.

### 4.4. Recommendations

Based on our experience illustrated here, we consider that combined amino acid PET–MRI is likely to improve the diagnostic accuracy of standard MRI in paediatric and TYA tumours in the following clinical situations:Surveillance imaging in low-grade gliomas.Discriminating tumour progression from treatment effects in high-grade gliomas.Assessment of early and end of treatment response in gliomas and intra-cranial germ cell tumours.Surgical and radiotherapy planning in diffuse gliomas.

Although the clinical situations presented do not necessarily require simultaneous acquisition in a PET–MRI scanner, the advantages, particularly in the paediatric population, are the requirement for only one examination, particularly if sedation or general anaesthetic is planned. Another advantage is the exact co-registration of PET and MRI, which is desirable post-surgery where anatomical distortion is likely to occur. However, there are several disadvantages in that PET–MRI is a lengthier examination process, at approximately 90 min acquisition, compared to MRI or PET alone. There is also likely to be decreased patient comfort due to smaller bore size causing claustrophobia and ear phones not being standard. Finally, the necessity of rigid head fixation for smaller head size can be a cause of increased imaging artefacts. However, next-generation PET–MRI patient scanners will hopefully improve patient comfort and realize the potential of a one-stop non-invasive imaging technique in neuro-oncology. It is important to stress that a one-off negative scan with PET–MRI during any phase of treatment does not equate to a complete metabolic response, and, hence, a baseline scan is mandatory. The ease of availability is a common limitation for both tracers— ^18^fluorocholine [FCho] and ^18^fluoro-L-phenylalanine ad will have to be factored in for clinical use.

#### 4.4.1. Premise of This Report

Is PET–MRI an effective tool for oncological imaging of primary brain tumours in children and TYA patients, especially in the delineation tumour extent, monitoring treatment response and post-treatment differentiation between tumour progression or recurrence versus treatment-related changes? The second question is, what is the optimal indication of PET–MRI imaging in brain tumours?

#### 4.4.2. What this Report Adds to Published Literature

Although conventional gadolinium-enhanced magnetic resonance imaging (MRI) is considered the primary imaging modality for patients with brain tumours, it has some limitations, primarily in differentiating tumour grade, its inability to provide metabolic or functional information of the biology of the tumour, and in distinguishing disease recurrence from post-therapy changes. This report corroborates the effectiveness as well as the limitations of PET–MRI in paediatric and TYA neuro-oncology in the following:
Facilitating diagnostic neuro-surgical intervention, i.e., the site for pre-operative biopsy planning and navigation in children and TYA patients;The reassurance that metabolically inert post-therapy residual lesions in patients with intracranial germ cell tumours represent non-viable tissue;Defining the tumour and target volumes for radiotherapy by combining metabolic and radiological information of tumour extent and infiltrative margins;The detection of tumour recurrence during clinical and radiological surveillance;In distinguishing tumour recurrence from treatment necrosis/therapy-related changes;That PET–MRI alone is insufficient in providing accurate delineation and diagnostic sensitivity of observed lesions, as not all histological subtypes of brain tumours take up the radiotracers.

#### 4.4.3. Published Evidence

There is limited published evidence of the use of PET–MRI as an imaging modality for diagnostic purposes, early treatment response assessment, end of treatment disease status or as an imaging tool for post-treatment radiological surveillance in a paediatric and TYA patients with brain tumours.

#### 4.4.4. Added Value of This Report

This review highlights the importance of a multi-parametric imaging and that PET–MRI unequivocally is a promising imaging tool for children and TYA patients with a variety of brain tumours. It allows for the diagnosis and monitoring of morphological and metabolic response and changes during therapy and is an effective imaging tool in detecting viable residual tumour at the end of treatment.

#### 4.4.5. Implication of All of the Available Evidence

Overall, the conventional cross-sectional MRI combined with PET is a complementary modality for assessment of tumour burden. Combining and matching both imaging modalities is significantly superior to either imaging modality alone in assessing the presence of viable residual tumour and indicating the radiotherapy target volume. Additionally, PET–MRI generates quantitative measures that may add significant value to conventional SUV image-derived measures.

## 5. Conclusions

While for primary tumour staging, MRI on its own is sufficient, the use of combined PET and MRI provides additional molecular and functional information for critical treatment decisions. This is highly relevant in clinical neuro-oncology, and the use of PET–MRI will increase in the years to come. Whilst mostly a research tool in paediatric/TYA neuro-oncology, there is enough evidence from the adult neuro-oncology literature to justify its clinical use. We have demonstrated a number of clinical situations where hybrid PET–MRI can improve diagnostic accuracy, and as improvements in PET–MRI scanners are made, the use in paediatric/TYA neuro-oncology is likely to increase.

## Figures and Tables

**Figure 1 jpm-10-00218-f001:**
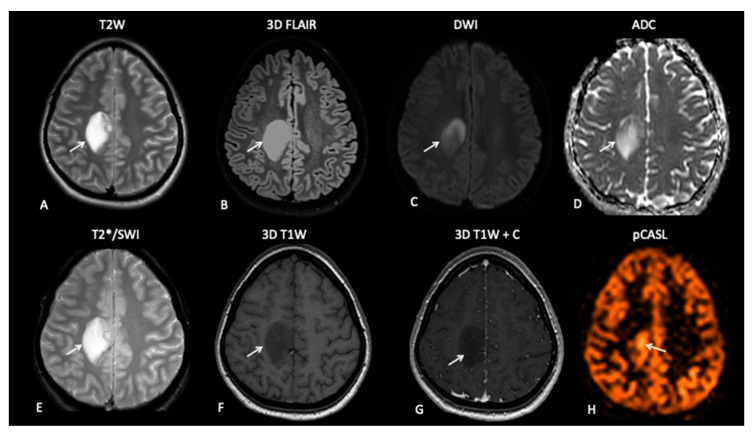
Nineteen-year-old female presenting with epilepsy. MRI demonstrates a well-defined T2W (**A**), FLAIR (**B**), DWI (**C**) and a hyperintense lesion with a medial border of restricted diffusion on an apparent diffusion coefficient (ADC) map (**D**). The lesion demonstrates no areas of susceptibility (**E**) and no pathological enhancement on pre- and post-contrast T1W images (**F**,**G**). The appearances are consistent with low-grade glioma, but pCASL perfusion-weighted imaging (PWI) (**H**) demonstrates medial increased perfusion.

**Figure 2 jpm-10-00218-f002:**
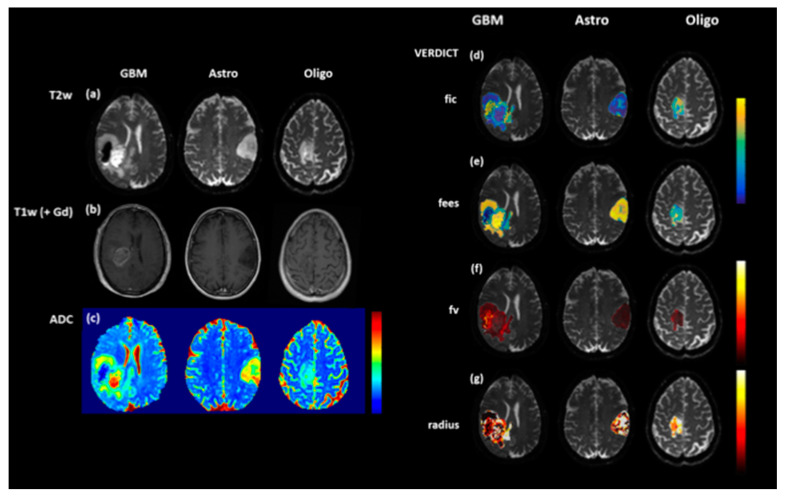
T2-weighted (T2W (**a**), T1-weighted (T1W) post contrast (**b**) and ADC maps (**c**) in a range of brain tumours: glioblastoma multiforme (GBM), diffuse astrocytoma (Astro) and oligodendroglioma (Oligo). Although the ADC map for the oligodendroglioma (**c**) is relatively uniform, the Vascular, Extracellular, and Restricted Diffusion for Cytometry in Tumours (VERDICT) parameter map fic (intracellular fraction) (**d**) demonstrates increased signal at the anterior margin, thought to be a marker of increased cell density. The other VERDICT parameter maps (**e**) fees (extracellular, extravascular fraction), (**f**), fv (vascular fraction) and (**g**) radius (cell radius) also demonstrate tumour heterogeneity which could potentially be used to guide biopsy.

**Figure 3 jpm-10-00218-f003:**
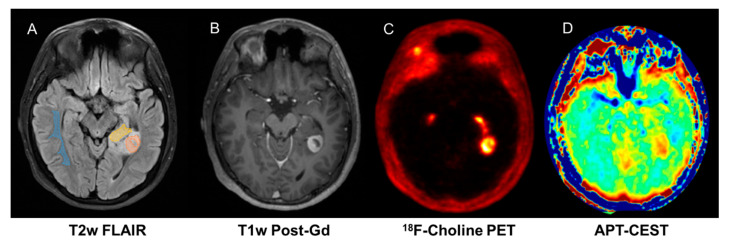
Fifteen-year-old male with pilocytic astrocytoma imaged on hybrid PET/MR which enables exact co-registration of FCho (**C**) with APT-CEST (**D**) showing correlation in enhancing ((**B**), T1 contrast) and non-enhancing tumour components. ((**A**), T2 FLAIR).

**Figure 4 jpm-10-00218-f004:**
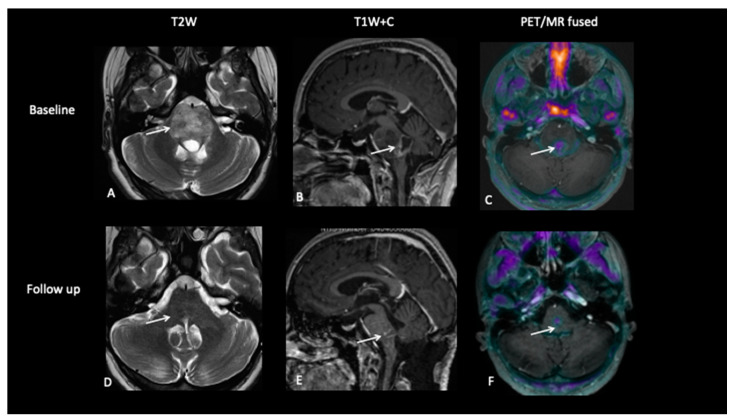
At baseline, there is a large pontine tumour (**A**) with a dorsally exophytic enhancing component (**B**), which shows FCho avidity on the PET/MR fused image (**C**). Follow-up imaging demonstrates residual dorsal enhancement (**D**,**E**), but FCho confirms no avidity (**F**) consistent with complete metabolic response.

**Figure 5 jpm-10-00218-f005:**
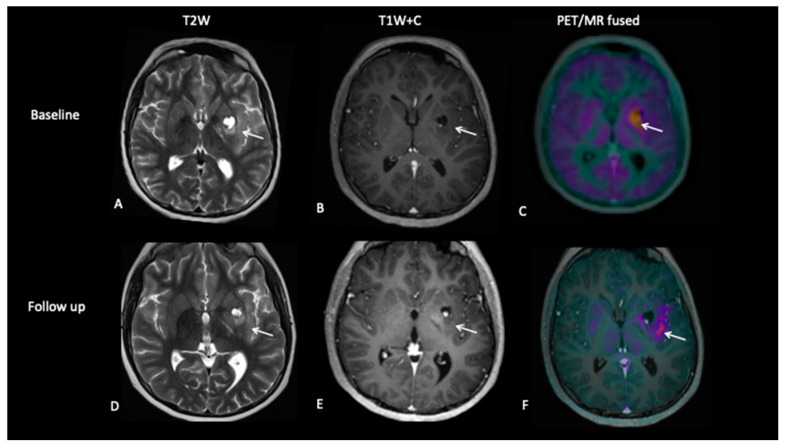
At baseline, there is non-enhancing tumour involving the left basal ganglia (**A**,**B**) with ^8^F-fluorocholine (FCho) and ^18^fluoro-L-phenylalanine (FDOPA) avidity (**C**). Follow-up imaging shows residual non-enhancing tumour (**D**,**E**) reported as stable disease, but FDOPA PET demonstrates reduced uptake (**F**) consistent with metabolic partial response.

**Figure 6 jpm-10-00218-f006:**
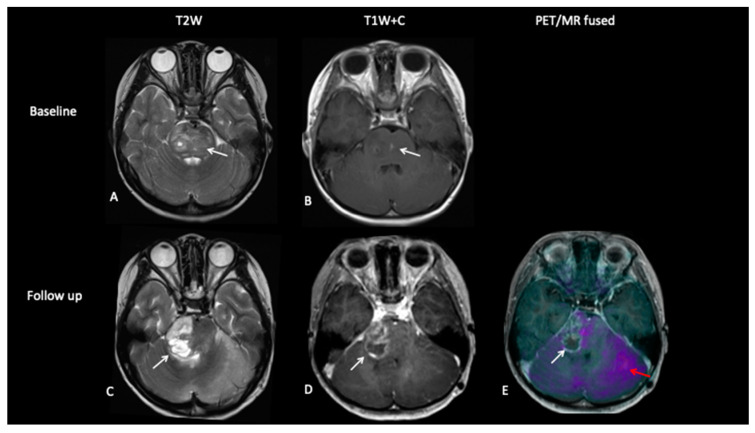
Baseline imaging demonstrates large pontine tumour (**A**) with focus of enhancement (**B**). Follow-up imaging demonstrated a large heterogenous right hemipontine mass (**C**,**D**) suspicious for tumour progression, but FDOPA (**E**) did not show increased uptake (white arrow) and was suggestive of pseudoprogression. However, FDOPA did show increased avidity in the left cerebellar hemisphere (red arrow) confirming non-enhancing tumour progression at that site.

**Figure 7 jpm-10-00218-f007:**
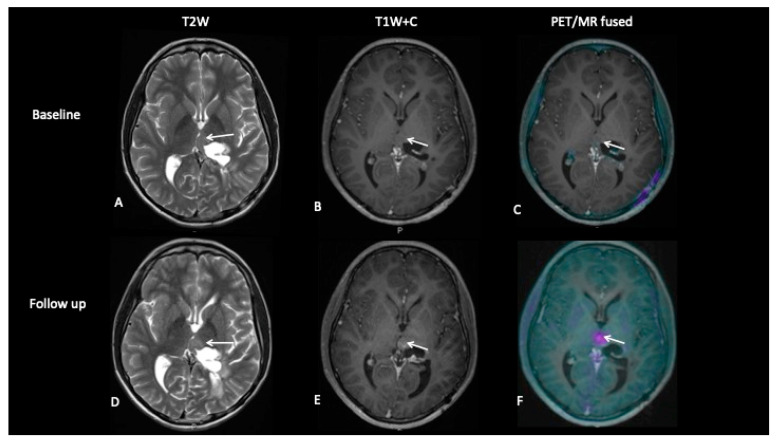
The baseline imaging post-surgery demonstrates residual non-enhancing tumour in the medial right thalamus (**A**–**C**). Follow-up imaging demonstrates the non-enhancing component (**D**) and mild enhancement of this component (**E**), uncertain for tumour progression or pseudoprogression, but the FDOPA (**F**) demonstrates increased uptake consistent with tumour progression.

**Figure 8 jpm-10-00218-f008:**
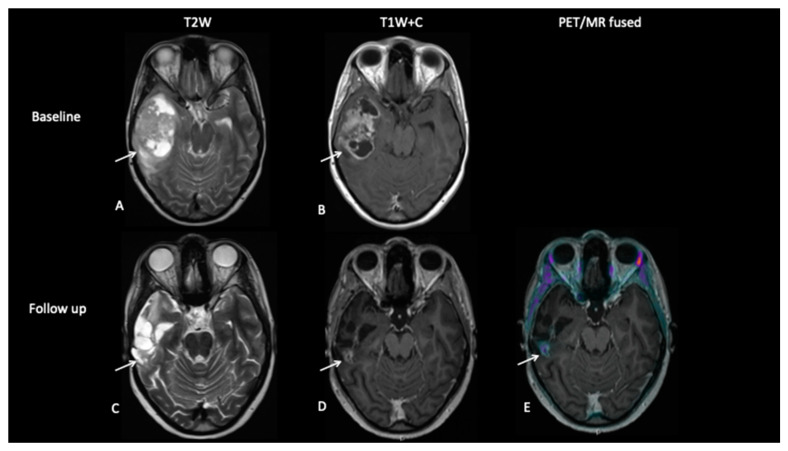
Baseline imaging demonstrates a large heterogenous right temporal lobe tumour (**A**) with avid enhancement (**B**). Follow-up imaging demonstrates the post-surgical resection on the T2W image (**C**) with nodular enhancement at the deep surgical margin (**D**), suspicious for residual tumour, but there is no increased uptake on FCho (**E**).

**Figure 9 jpm-10-00218-f009:**
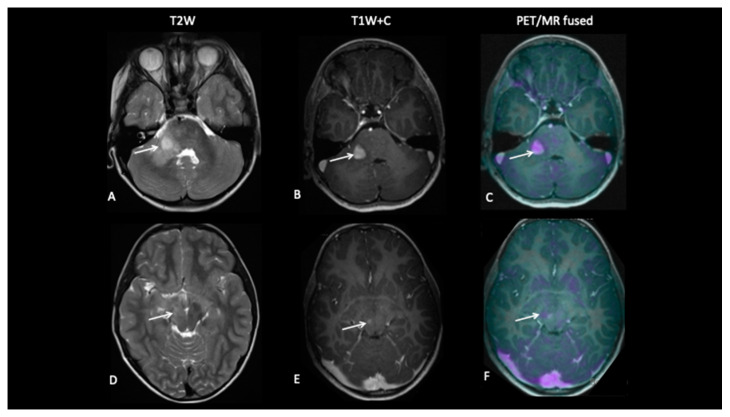
A mid-treatment scan demonstrates an enlarging right cerebellar hemisphere lesion (**A**,**B**) thought to reflect pseudoprogression, but FDOPA demonstrates increased uptake (**C**) and identifies a further lesion in the midbrain (**F**) not obviously apparent on the T2W image (**D**) or T1W post-contrast image (**E**), which is consistent with multifocal tumour progression.

**Figure 10 jpm-10-00218-f010:**
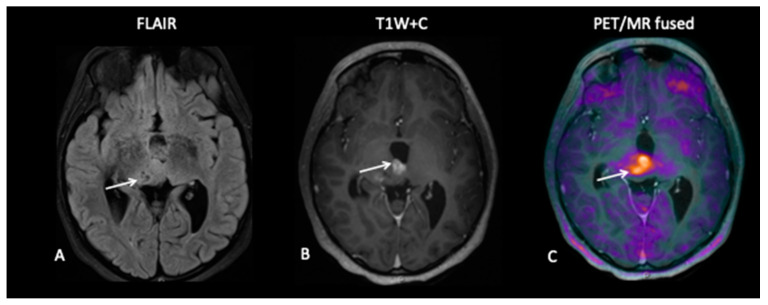
The MRI study demonstrates a large area of signal abnormality in the right thalamus (**A**) associated with a medial enhancing nodule (**B**). However, FDOPA demonstrates increased uptake in the non-enhancing component in addition to the enhancing component (**C**), confirming metabolically active non-enhancing disease in the right thalamus.

**Figure 11 jpm-10-00218-f011:**
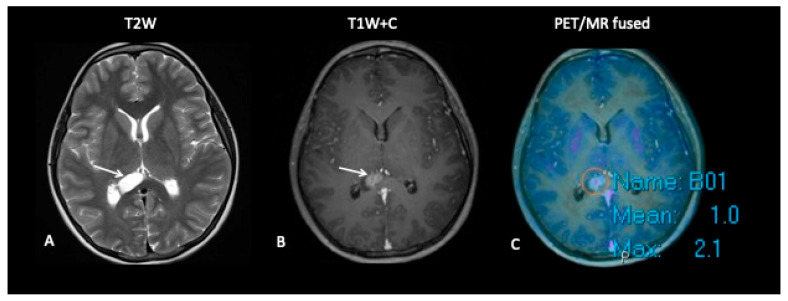
MRI demonstrates an expansile lesion in the right aspect of the splenium (**A**), which demonstrates heterogenous enhancement (**B**), suspicious for malignant transformation. However, FDOPA (**C**) did not demonstrate increased uptake, and the lesion was considered low grade.

**Figure 12 jpm-10-00218-f012:**
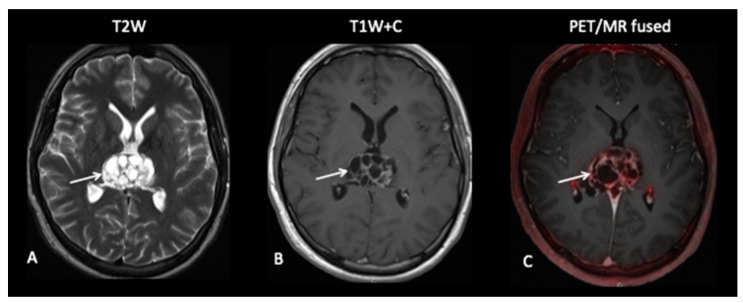
MRI demonstrated a stable residual cystic large pineal region tumour (**A**) with enhancement of the internal septations (**B**) reported as stable disease. However, 18F-fluoroethylcholine (FEC) (**C**) showed increased avidity consistent with metabolically active disease.

**Figure 13 jpm-10-00218-f013:**
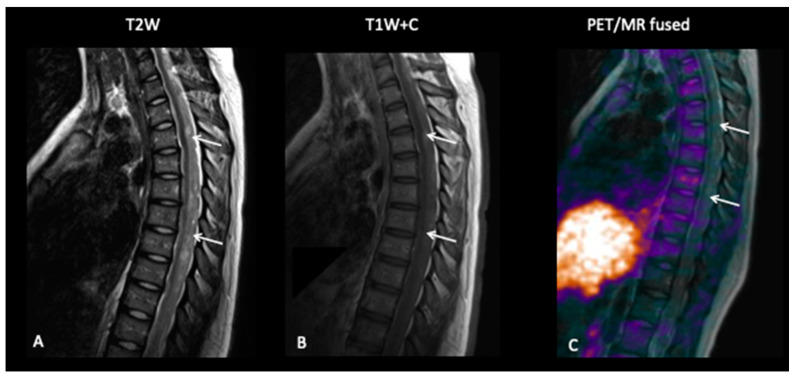
Spinal MRI (**A**,**B**) demonstrates extensive intraspinal-enhancing soft tissue in keeping with leptomeningeal metastatic disease (arrows point out the leptomeningeal disease). However, the FCho PET (**C**) did not demonstrate avidity of the leptomeningeal disease (arrows).

**Table 1 jpm-10-00218-t001:** Neuro-Oncology MRI protocol sequence parameters.

		TR	TE	FA	FOV	ACQ Matrix	Reconstructed Voxel	Slices
**DWI**	SE-EPI	2902	95	90	230 × 230	152 × 106	0.9 × 0.9 × 5	22
**FLAIR**	IR-TSE	11000	125	90/120	230 × 182	352 × 186	0.45 × 0.45 × 5	29
**T2WSE**	TSE	3000	80	90/120	230 × 184	400 × 255	0.45 × 0.45 × 5	24
**T2WFFE**	FFE	786	15	18	230 × 183	256 × 164	0.45 × 0.45 × 5	24
**pCASL**	GEPI	4571	15	90	240 × 240	64 × 64	3 × 3 × 5	20

Note: DWI = diffusion weighted imaging, FLAIR = fluid attenuated inversion recovery, T2WSE = T2-weighted spin echo, T2WFFE = T2-weighted fast field echo, pCASL = pseudo-continuous arterial spin labelling, TR = relaxation time, TE = echo time, FA = flip angle, FOV = field of view, ACQ matrix = acquisition matrix.

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
