# Peer review of "Hybrid PET–MRI Imaging in Paediatric and TYA Brain Tumours: Clinical Applications and Challenges"

_jpm, 2020, doi:10.3390/jpm10040218_

Round 1
Reviewer 1 Report
In this manuscript, Shankar et. al. presented a literature review and a brief case series of PET-MR imaging of pediatric and young adult brain tumors. The focus of this manuscript is timely and innovative. While I enjoyed the topic of this study, I am confused on the overall scope and structure of the manuscript as written.
In order to evaluate this manuscript more carefully, the authors need to decide if they are presenting a literature review or a case series. The abstract indicates that the manuscript is a case series, whereby the authors critically evaluate of 60 cases. However, the manuscript is largely a literature review with images a brief summary of only 10 patients. The data is not collated, nor is there any analysis done on their own data to support the conclusion in the abstract ‘Combined PET/MRI shows promise for improved diagnostic assessment of pediatric and TYA brain tumors’.
Author Response
Response: Four points have been raised and we clarify each point below:
- Patient number: While PET scan for brain tumour diagnostics and response assessment have been done in approximately 50-60 children and adolescents, not all were done on the hybrid PET MRI scanner. The earlier scans were done separately in two scanners [PET and MRI] and the images were superimposed/fused. Secondly, the group of patients included in this report all underwent scanning in our hybrid PET MRI scanner. We have selected patients with different histological subtypes of brain tumours to illustrate our point of the utility of using PET MRI in brain tumour patients. This is not a critical evaluation of 60 patients. This has been clarified in the abstract
- Literature review: This is not a literature review as such as there is not much in published literature on hybrid PET MRI in children and adolescents with brain tumours. We have cited some published information to support our premise of the value of PET MRI in children and adolescents with brain tumours. We have published 4 papers on PET MRI in children with brain tumours – references are in this report. However, our publications focus on individual subtypes – gliomas, intra cranial germ cell tumours etc. and here we present a smorgasbord of information on the different subtypes of brain tumours as a single report
- Collation of data: We have restructured the draft and we hope the data now is clearer.
- Analysis of our own data: We have used the information on 10 patients as supportive evidence of the utility of hybrid PET MRI. The last 2 pages in this submission clarify the purpose of this report.
Reviewer 2 Report
The manuscript titled “Hybrid PET-MRI imaging in Paediatric and TYA brain tumours: clinical applications and challenges”, unfortunately, is highly disorganized, therefore hard to read. It lacks clear structure. The reference list does not correspond to journal roles (list is inconsistent).
Author Response
1. We have numbered the headings and sub-headings in the manuscript to make the structure clearer.
2. We have also reformatted the references.
Reviewer 3 Report
PET-MRI is advantageous for quantitative evaluation of tumors. However, could the authors discuss more about its capability to stage tumors?
Also, are there any potential shortcomings of the two PET agents used? How will their permeability to different tumors and retention time affect the diagnosis accuracy?
Author Response
PET-MRI is advantageous for quantitative evaluation of tumours. However, could the authors discuss more about its capability to stage tumours?
For primary tumour staging MRI on its own is sufficient. At present there is no large prospective data to support the combined use of PET-MRI to stage brain tumours (this has been stated under the title of Hybrid PET/MRI). We have clarified this again in the discussion
Also, are there any potential shortcomings of the two PET agents used? How will their permeability to different tumours and retention time affect the diagnosis accuracy?
- Ease of availability of the two tracer is one common limitation for 18F-fluorocholine (FCho) and 18fluoro-L-phenylalanine (FDOPA). - This has been clarified in the discussion
- Also a one off negative scan during any phase of treatment does not equate to complete metabolic response, a baseline scan is mandatory (this is mentioned in Discussion and has been clarified again).
- Increased permeability and prolonged retention time will improve detection of any residual tumour
Round 2
Reviewer 2 Report
In this article, entitled ”Hybrid PET-MRI imaging in Paediatric and TYA brain tumours: clinical applications and challenges”, Ananth Shankar and colleagues 1) analyse literature focused on PET/MRI giving “an overview of the current imaging techniques used in clinical practice”, and 2) present 10 clinical cases from their practical experience. This is indeed important research aiming in order to improve methods for the treatment of brain tumours and giving the insight in the neuro-oncology practice.
However, the difficulty in this reviewer's opinion and specific comments are listed below:
- The paper seems not to be carefully prepared and checked before the submitting.
- The manuscript suffers from a number of inconsistencies both in the “review section” and “an experimental section”.
- Together 67 references are not enough for “review” article; references 1-16 are with missing pages, volumes, etc. There are serious confusions in the text regarding figures in “an experimental section”, statistical analysis is lacking.
- Line 211-215, in the legend of figure1, “C” and its explanation is missing.
- Line 258, the legend is poor in the figure 2 (for example, “GBM”, ”Astro”, “Oligo”, “fic” , “fees”, “fv”, “radius” meanings are not explained, “E”, “F”, “G”and its explanations are missing).
- Line 351, “A”, “B” and its explanations are missing in the legend of figure 3.
- Line 367, the authors mention 4 specific tumour types, however, only short description is given with any references in the “review” section (high grade gliomas), similarly in the line 526 (intra cranial germ cell tumour) and in the line 556 (primitive neuro-ectodermal tumours). No any characterization/references for “low grade gliomas” (line 470).
- Line 389, an explanation for “C” (“C” is missing) in the legend of figure 4.
- Line 444, an explanation for “D” (“D” is missing) in the legend of figure 7.
- Line 466, an explanation for “C” (“C” is missing) in the legend of figure 8. The legends of all figures could be more informative for readers (for example, “follow up” explanation is missing).
- Line 472, what means “1” in the end of the first sentence.
- Line 486, an explanation for “D” and “E” (“D” and “E” are missing) in the legend of figure 9.
- Line 496, an explanation for “C” (“C” is missing) in the legend of figure 10.
- Line 574, no explanations for arrows showed in the figure 13.
- Discussion is very poor (lines 580-589 without any references, paragraph 591-595 without any references and discussion: discussion must take into the account research of other researchers on this topic. This has not been done. In the paragraph 598-603, the authors used only self-citation, including abstract (Multiparametric 18F-Choline PET/MR in childhood and teenage-young adult gliomas: assessment of suspected disease progression, Neuro-Oncology, Volume 21, Issue Supplement_4, October 2019, Page iv5). Following paragraphs show only few references/data comparison/analysis.
- Line 694-702, it is not clear why authors describe “review questions”, “what review adds” etc. after the “conclusions”.
Finally, based on 10 cases, the evaluation is not enough for claiming “combined PET/MRI shows promise for improved diagnostic assessment in paediatric and TYA brain tumours”.
Author Response
- The paper seems not to be carefully prepared and checked before the submitting – This has been done very thoroughly and all errors/ omissions corrected
- The manuscript suffers from a number of inconsistencies both in the “review section” and “an experimental section” – This has been addressed
- Together 67 references are not enough for “review” article; references 1-16 are with missing pages, volumes, etc. There are serious confusions in the text regarding figures in “an experimental section”, statistical analysis is lacking. References have been re structured and all missing information added
- Line 211-215, in the legend of figure1, “C” and its explanation is missing – This has been done
- Line 258, the legend is poor in the figure 2 (for example, “GBM”, ”Astro”, “Oligo”, “fic” , “fees”, “fv”, “radius” meanings are not explained, “E”, “F”, “G”and its explanations are missing) – This has now been done.
- Line 351, “A”, “B” and its explanations are missing in the legend of figure 3. This has been corrected
- Line 367, the authors mention 4 specific tumour types, however, only short description is given with any references in the “review” section (high grade gliomas), similarly in the line 526 (intra cranial germ cell tumour) and in the line 556 (primitive neuro-ectodermal tumours). No any characterization/references for “low grade gliomas” (line 470). We sub-categorized this section to address the 4 types of tumours according to the imaging that follows each sub category. We do not feel that references are required here. However, we have further clarified this in the revised manuscript.
- Line 389, an explanation for “C” (“C” is missing) in the legend of figure 4. This has been done.
- Line 444, an explanation for “D” (“D” is missing) in the legend of figure 7. This has been done
- Line 466, an explanation for “C” (“C” is missing) in the legend of figure 8. The legends of all figures could be more informative for readers (for example, “follow up” explanation is missing). This has been done
- Line 472, what means “1” in the end of the first sentence. This has been amended. It is NF1 or neurofibromatosis type 1
- Line 486, an explanation for “D” and “E” (“D” and “E” are missing) in the legend of figure 9. This has been done
- Line 496, an explanation for “C” (“C” is missing) in the legend of figure 10. This has been provided.
- Line 574, no explanations for arrows showed in the figure 13. This has been done.
- Discussion is very poor (lines 580-589 without any references, paragraph 591-595 without any references and discussion: discussion must take into the account research of other researchers on this topic. This has not been done. In the paragraph 598-603, the authors used only self-citation, including abstract (Multiparametric 18F-Choline PET/MR in childhood and teenage-young adult gliomas: assessment of suspected disease progression, Neuro-Oncology, Volume 21, Issue Supplement_4, October 2019, Page iv5). Following paragraphs show only few references/data comparison/analysis. This has now been restructured and discussion expanded.
- Line 694-702, it is not clear why authors describe “review questions”, “what review adds” etc. after the “conclusions” This has been corrected and the paper carefully re-structured to reflect the comment of the reviewer.
Finally, based on 10 cases, the evaluation is not enough for claiming “combined PET/MRI shows promise for improved diagnostic assessment in paediatric and TYA brain tumours”. We believe that the statement “ “combined PET/MRI shows promise for improved diagnostic assessment in paediatric and TYA brain tumours” is true as the word promise denotes potential. However, we have re-phrased the manuscript to reflect this in the discussion, the premise of this report, the drawbacks of PET MRI – in the paragraph “what this report adds to published literature , added value of this report and recommendations the conclusion.
